



# Remote sensing in an index-based insurance design for hedging economic impacts on rice cultivation

Omar Roberto Valverde-Arias[1], Paloma Esteve[12], Ana María Tarquis[13], and Alberto Garrido[12]

[1]CEIGRAM, ETSIAAB, Universidad Politécnica de Madrid, Madrid, 28040, Spain

[2]Dpto. de Economía Agraria, Estadística y Gestión de Empresas, ETSIAAB, Universidad Politécnica de Madrid, 28040 Madrid, Spain

[3]Grupo de Sistemas Complejos, ETSIAAB, Universidad Politécnica de Madrid, 28040 Madrid, Spain

*Correspond to:* Omar Valverde (omar.valverde@upm.es)

**Abstract:**

Rice production in Ecuador is steadily affected by extreme climatic events that make it difficult for farmers to cope with production risk, threatening rural livelihoods and food security in the country. Developing agricultural insurance is a policy option that has gained traction in the last decade. Index-based agricultural insurance has become a promising alternative that allows insurance companies to ascertain and quantify losses without verifying a catastrophic event in situ, lowering operative costs and easing implementation. But its development can be hindered by basis risk, which occurs when real losses in farms do not fit accurately with the selected index. Avoiding basis risk requires assessing the variability within the insurance application area and considering it for representative index selection. In this context, we have designed an index-based insurance that uses a vegetation index (NDVI) as indicator of drought and flood impact on rice in Babahoyo canton (Ecuador). Babahoyo was divided in two Agro-ecological Homogeneous Zones to account for variability, and two NDVI threshold values were defined to consider, first, the event impact on crop (physiologic threshold), and, second, its impact on gross margin (economic threshold). This design allows us to set-up accurate insurance premiums and compensations that fit the particular conditions of each AHZ, reducing basis risk.

## 1 Introduction

Rice cropping area in Ecuador is witnessing a reduction trend along recent years (FAO, 2018). From an average cultivated area around 400,000 ha between 2005 and 2015, annual average decreased to 385,039 ha in 2016 and to 370,406 ha in 2017, falling considerably to 301,853 ha in 2018 (Aguilar et al., 2015, 2018; INEC, 2018; Montaño, 2005). Such a downward trend rises Government's concern as rice production plays an important role in Ecuadorian food security (Pinstrup-Andersen, 2009) and it is central to rural livelihoods in certain areas of the country. Daily rice consumption per person is 115g (Montaño, 2005), which represent currently an annual demand of 714,000 tonnes. Additionally, rice production in Ecuador offers employment to 22% of the economically active population, involving around 140,000 families. For these reasons, Ecuadorian government supports rice producers through technical advice, subsidized inputs, credit lines for farm modernisation, and minimum support prices (Eymond and Santos, 2013). However, these supporting mechanisms have not prevented efficiently the gradual reduction of rice cropping area, being necessary to adopt additional measures that support stability of farmers' revenues.

FAO and UN-Habitat, (2010) reported the 29 most important disasters in Ecuador in the last twenty years, 59% of which had climatic origin. Additionally, the most common extreme climatic events in Ecuador are flood and drought according to the





Centre for Research on the Epidemiology of Disasters-CRED (2015). Sivakumar et al., (2005) mentioned that extreme climatic
events have increased both in frequency and intensity, making it more difficult for farmers to maintain their crop productions
(Cai et al., 2014; Isch, 2011). These climatic phenomena, which are further accentuated by climate change, are key drivers of
economic losses that hit especially Tropic's small rice farmers (Harvey et al., 2014), and are one of the main reasons behind
rice cropping area loss in Ecuador (Eymond and Santos, 2013; Poveda and Andrade, 2013). For instance, the 2012-winter's
impact census over agriculture (MAGAP, 2012), showed that from 140,000 cultivated hectares analysed, 56,562 ha were
entirely destroyed by flood and 24,103 ha were partially damaged by the same event. In this context, risk management
mechanisms, such as agricultural insurance, can importantly contribute to reduce rice producers' vulnerability and to protect
them against the economic losses driven by climatic extremes.
Agricultural insurance is an effective tool for transferring production risk from farmers to other entities. It allows farmers to
meet their credit obligations and minimize the effect of extreme climatic events on their revenue (Xu and Liao, 2014).
Moreover, agricultural insurance contributes to maintain farmers in the agricultural business, improve their resilience and
preserve food security (Bullock et al., 2017; Patt et al., 2009). In pursuit of these goals, Ecuador started to implement in 2010
conventional insurance through the AgroSeguro system that includes a 60% subsidy of insurance's premium cost (MAG,
2018). This is a multi-peril insurance system that covers some crops, including rice, requiring an in situ verification in case of
disaster occurrence. Under the coverage of this insurance, in case of a generalized extreme event, the insurance company's in
situ verification capacity could be exceeded, delaying payouts, and some remote regions could be uncovered. Moreover,
(Medina, 2017) suggest that conventional insurance in Ecuador may be inefficient due to asymmetric information that may
increase adverse selection and moral hazard. Therefore, even if current AgroSeguro insurance system has importantly
supported farmers along the last decade, it is important for the Ecuadorian Government to step forward to the next level in
agricultural insurance field to expand the insurance coverage and reduce transaction costs resulting in lower premium prices
and a more efficient system.
Among different types of agricultural insurance schemes, index-based insurance (IBI) is a promising tool to provide coverage
to large agricultural areas around the world (Mobarak and Rosenzweig, 2013), based on the use of a highly losses-correlated
index that avoids the need for field losses verification (Carter et al., 2011). The use of such an index as trigger for indemnity
payments reduces significantly the costs for the insurance company in relation to losses verification and payment procedure,
and reduces fraud, moral hazard and adverse selection (Barnett and Mahul, 2007; de Leeuw et al., 2014) that are frequent
drawbacks of conventional insurance. IBI has been underlined as a feasible and efficient risk management tool (Jensen and
Barrett, 2017; Jensen et al., 2018; Takahashi et al., 2016), and several studies demonstrated its successful implementation
using weather and vegetation index among small and medium farmers in developing countries (Mcintosh et al., 2013; Mude
et al., 2009, among others) that can benefit from lower insurance premiums due to lower implementation costs. In this regard,
IBI represents an alternative to conventional insurance in Ecuador, which could be applied by insurance companies and the
Government to satisfy the risk management needs of rice producers.
However, the technical, economic and administrative hurdles are significant. A major problem that may arise in the
implementation of the IBI is the lack of proper correlation between the index and the losses experienced by farmers in the
index influence area (IIA), which is the area for which a defined index is representative (Elabed et al., 2013). This problem,
known as basis risk, occurs when some farmers from the pool of insured agents do not receive any compensation even
experimenting losses, and some others not being affected are indemnified (Clarke, 2016; Hellmuth et al., 2009). To avoid this,
IBI can only be applied over spatially homogeneous areas because its main principle is based on the use of a single index over
the IIA. Nevertheless, these conditions of homogeneity are rarely found because agriculture is practiced in heterogeneous


areas. To keep basis risk in non-significant levels, index selection and analysis may be crucial, very especially with respect to
the way variability within the IIA could influence index values.
Among the indexes used in IBI design, several authors (e.g. Jensen et al., 2018; Rao, 2010) underlined vegetation indexes such
as those based on the Normalized Difference Vegetation Index (NDVI), as options that reduce basis risk and provide reasonably
accurate loss estimations, and that can significantly profit from recent advances in remote sensing, geographical information
systems, and satellite and drone imagery among others. In line with this, in this research we aim to design an IBI based on
NDVI for rice crop in Ecuador that covers farmers against drought and flood events, accounting for variability within the IIA.
For this, we build upon previous work developed by Arias et al., (2018) for the rice-producing coastal region of Ecuador that
identified agro-ecological homogeneous zones (AHZ), based on topographic, soil, and climatic characteristics using principal
components and hierarchical cluster analysis. Within that area, in the Babahoyo canton, two AHZs (*f7 and f15*) were located
and their influence over the NDVI in rice cultivation was found significant (Valverde-Arias et al., 2019). For the IBI design,
two thresholds in the NDVI values will be defined. The physiologic threshold evidences the occurrence of an extreme climatic
event and its impact over rice-crop yield. While, the economic threshold is reached when a moderate climatic event occurs,
and its impact over the rice-crop yield is not so deep, letting farmers at least to cover the production costs. For these thresholds,
two scenarios are contemplated; the first one considers a differentiated production cost for each AHZ. The second scenario
uses the same average production cost for both zones *f7* and *f15*. Then, the damage compensation and the premium cost are
calculated for each threshold considering the two scenarios and the AHZs.

## 2    Materials and methods

This section presents the data and methods followed for the development of an IBI. Starting with a description of the study
area location (section 2.1) and data used (section 2.2), section 2.3 explains how we assessed the significance of AHZs impact
on NDVI. Then, we estimate the functional relationship between NDVI and rice yield (section 2.4), determine the NDVI
thresholds (section 2.5), and we assess risk in each AHZ (section 2.6). Finally, for the design of the IBI contract, in section 2.7
we explain first how indemnities are calculated and how the insurance premiums are estimated considering different zones and
different coverages.

### 2.1    Location of study area

Rice production in Ecuador concentrates in the coastal area of the country, very especially in the provinces of Guayas and Los
Rios (55% and 37% of rice cropping area respectively during the rainy season). This study focuses on rice cultivation area in
the Babahoyo Canton, which is one of the main rice producer areas in Los Rios Province (Fig. 1). 84% of the rural population
of Babahoyo is involved in agriculture, being rice the main crop in the region with 46,556 ha that represent 45% of the total
cultivated area in this canton (IEE, 2009; MAGAP, 2014). The location of Babahoyo in an extensive plain of the Ecuadorian
coastal region makes it very vulnerable to flood, and as Valverde-Arias et al., (2018) mentioned in their study this canton is
also susceptible to droughts. Therefore, given the importance of rice production in the region's economy and its vulnerability
to hazardous climatic events, designing and implementing an IBI that accounts for variability within the area and that provides
accurate premium prices and indemnities may importantly contribute to rice producers' welfare and stability.

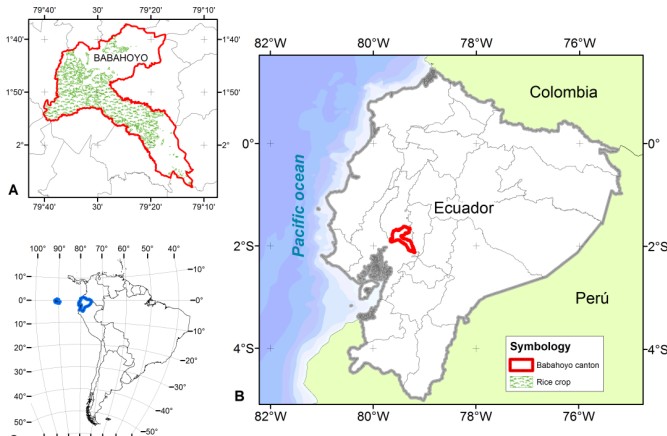


**Figure 1: (A) Babahoyo canton with rice crop coverage, (B) location of Babahoyo canton in Ecuador, and (C) location of Ecuador in South America**

### 2.2    Cartographic Data

#### 2.2.1    Agro-ecological Homogeneous zones map

In this research we build on the AHZ map generated for the Ecuadorian Coastal region in the study of Arias et al., (2018) that includes our study area (Babahoyo canton). In this map, portions of land with similar agro-ecological characteristics were grouped in homogeneous zones (AHZs) using a statistical method of principal component analysis and a hierarchical cluster analysis. According to the map, there are eleven AHZs in Babahoyo, from which seven include rice crop cultivation land. Two of these seven AHZ, *f7* and *f15*, were selected for the study as they account for more than 90% of the total rice-cultivated area in Babahoyo (Valverde-Arias et al., 2019) (see Fig. 2).

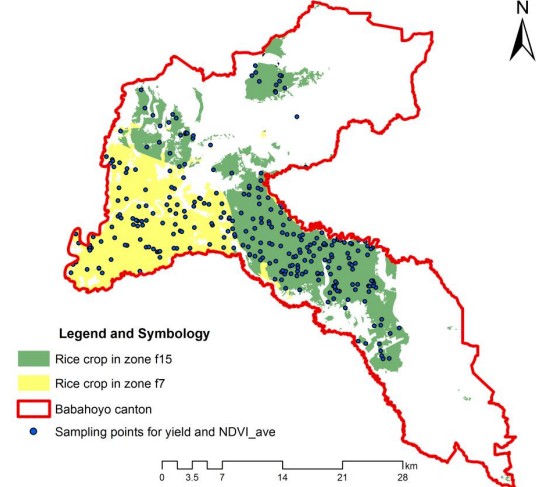


**Figure 2: Agro-ecological homogeneous zones *f7* and *f15* over rice cultivation area with yield observations in Babahoyo canton**





**2.2.2    Data from satellite imagery**
Satellite imaginary data were obtained from the MODIS MOD13Q1V6 product, which has the following characteristics
(NASA LP DAAC, 2015), see Table 1.

**Table 1.** Technical characteristics of MODIS imagery set

| Characteristic | Description |
| --- | --- |
| Temporal Granularity | 16-day |
| Temporal Extent | 2001-2017 |
| Spatial Extent | Ecuador |
| Coordinate System | Projected to Universal Transverse Mercator |
| Datum | WGS 1984 Zone 17 S |
| File Format | HDF-EOS |
| Geographic Dimensions | 1200 km x 1200 km |
| Number of Science Dataset (SDS) Layers | 12 |
| Rows/Columns | 4800 rows x 4800 cols |
| Pixel Size | 250 m |

Adapted from (Didan, 2015)
The imagery covers the rice cycle during the rainy season (January to May). There is one image for each 16-day period from
2001 to 2017, which makes 170 images in total (17 years x 5 months x two per month). The rice crop cycle in Ecuador takes
120 days. The sowing date starts around January 15[th], and sometimes it is delayed depending on the onset of the precipitations.
The downloaded imagery have a hierarchical data format (HDF), which is a multilayer file (twelve layers) (Didan, 2015);
however, we used only the layer Hdf:0 that corresponds to NDVI values.
**2.3    Statistical analysis**
NDVI values over rice along its crop cycle were analysed for the period 2001 to 2017. NDVI_ave is the average of all NDVI
measures of rice crop cycle (January to May) for each observation point. We sampled 30% of the total pixels of rice crop in
Babahoyo canton resulting in 31,756 observations: 13,498 in AHZ *f7* and 18,258 in AHZ *f15*.
Descriptive statistics were applied to the NDVI_ave data set, including the normality test of Kolmogorov-Smirnov, which is
recommended for more than 50 observations (Razali and Wah, 2011). If the data set fits a normal distribution, an analysis of
variance ANOVA will be applied for comparing means of two variability factors (zones and years). Otherwise, we will
determine which distribution this data set fits, and the test of Kruskal-Wallis for comparing median of AHZs and years will be
used. If significant differences are found among years, the Least Significance Difference (LSD) multiple rank test for means
(Williams and Abdi, 2010) or the Bonferroni test for medians will be applied. Years that are not significantly different will be
grouped into five categories based on NDVI_ave values: very low, low, normal, high, and very high years.
**2.4    Rice-Yield estimation through NDVI_ave**
According to Huang et al., (2013) remote sensing products can be used for generating yield estimation models that do not
require variables, as crop management or fertilizer applications. Robust results are obtained in rice-yield prediction even at
province level. Quarmby et al., (1993) mentioned that rice and maize yields could be estimated accurately by a simple linear
regression between NDVI and yield; in addition, Son et al., (2014) suggested that the use of multi-temporal NDVI data for
estimating rice-yield in large scale should be a possible and accurate alternative. In this research, we used the normal
distribution Eq. (1) for estimating rice yield from NDVI_ave values, quantifying in this way the economic losses in rice





cultivation caused by extreme climatic events. The estimation of rice yield was based on the relationship with the NDVI_ave
and the crop state.
$$Y = \frac{1}{\sigma\sqrt{2\pi}}\, e^{-\frac{(X-\mu)^2}{2\sigma^2}}$$ (1)
Where:
$\sigma$ = Standard deviation
$\sigma^2$ = Variance
$X$ = Independent variable (NDVI_ave)
$Y$ = Dependent variable (estimated rice yield)
$\mu$ = Arithmetical Mean of NDVI_ave in years 2016 and 2017
The General Coordination of the National Information System (CGSIN-acronym in Spanish-) of Ecuadorian Agricultural and
Livestock Ministry (MAG) has conducted a rice-yield estimation project since 2014 when it began sampling yields across
mapped rice areas. Thus, 369 georeferenced rice -yield observations (t/ha) were available for 2014-2017 rainfed cycles
(January to May) in the study area over AHZs *f7* and *f15* (see, Fig. 2). Therefore, we used these rice yield observations with
their corresponding spatial and temporal NDVI_ave values for obtaining the parameters included in Eq. (1) (Valverde-Arias
et al., 2019). The robustness of this model was evaluated through the RMSE (%) and R-squared coefficient.
**2.5   Thresholds determination**
There are three different levels of rice crop loss impacts, caused by drought and flood, that should be evaluated based on the
vegetation index selected. In the first one, catastrophic impact, the crop is acutely affected and the farmers cannot recover any
part of their investment. In the second level, physiological impact, the crops are strongly affected but farmers can recover part
of their investment. Finally, economic impact, the crop loss impact still allows farmers to recover their investment to break-
even or have a null profit. To differentiate in these three levels two NDVI_ave thresholds are needed.
According to LSD and Bonferroni multiple range tests, years with the lowest NDVI_ave means and medians are selected as
the more representative of physiological threshold. Then, we contrast if these years have been actually affected by flood or
drought through the climatic application of National Oceanic and Atmospheric Administration (NOAA, 2018). Finally, we
verified that these thresholds correspond to the reality comparing the estimated yield obtained using the NDVI_ave thresholds
with the expected yields in each AHZ and *cantonal* (at Babahoyo canton level) in normal years.
For the economic threshold, we set an NDVI_ave value that let farmers cover at least their production cost. Thus, we considered
the sale price at farm gate for a tonne of rice and the production cost in two scenarios: scenario 1 (when we consider
differentiated production cost for AHZs *f7* and *f15*) and scenario 2 (non-differentiated production cost for AHZs).
According to CGSIN, there are officially three different rice-crop production systems in Ecuador for rainfed agriculture and
two for irrigated agriculture in 2017. Each of them has different production costs as shown in Table 2, and they depend on the
level of farm modernization and whether they are rainfed or irrigated.






**Table 2. Official production cost of different rice-production systems in Ecuador in 2017**

| Rice cultivation production cost (USD/ha) | | | | |
|---|---|---|---|---|
| Rainfed production system | | | Irrigated production system | |
| Non-technical | Semi-technical | Technical | Semi-technical | Technical |
| 1022.0 | 1629.7 | 1955.9 | 1631.0 | 1997.4 |

Source: MAG, (2017)
Since we assessed rice production during rainy season (January-May), irrigation is not required in normal conditions. For this
reason, we use production costs of rainfed agro-systems. Among rainfed production systems, we chose the non-technical and
semi-technical systems, which are more exposed to suffer the impacts of extreme climatic events, and therefore are the ones
that should adopt insurance. We assigned to *f7* the production cost of a non-technical production system (1022 USD/ha) and
for *f15* the cost of semi-technical production system (1629 USD/ha) for the scenario one (see Table 2), as according to
Valverde-Arias et al. (2019) *f15* has an expected yield higher than *f7*'s yield in regular years that could be explained by *f15*'s
better soil conditions and to a more technical production system than in *f7*. Then, when we do not consider AHZs i.e., at
*cantonal* level, we used a weighted average production cost of these two systems (1259 USD/ha). In scenario two, i.e. when
similar costs are assumed for both AHZ, we used the weighted average (1259 USD/ha) for all the cases (*f7*, *f15*, and *cantonal*).
**2.6    Risk assessment in AHZs**
Once, we found the distribution that fits our data for each AHZ and *cantonal*, we simulated through these distributions a
determined number of NDVI_ave values. Then, we compared the frequency of observed NDVI_ave values with the estimated
ones. The basis risk of the estimation was evaluated through the Adjusted R-squared coefficient (Vedenov and Barnett, 2004).
Lastly, we calculated the proportion of positive events, that is, the number of events equal or under each threshold (physiologic
and economic) for each estimated distribution (*f7*, *f15* and *cantonal*). Finally, we tested whether these proportions of *f7* and
*f15* are significantly different from each other or not. This analysis was performed through the Z- test of two independent
proportions. It consists in contrasting if these two proportions which came from two different populations are equal (Pardo et
al., 1998; Polasek, 2013).
1.  Hypothesis:

$H_0: p_1 = p_2; \quad H_1: p_1 \neq p_2$

2.  Postulation: the studied variable (NDVI_ave) is dichotomous (below/equal or above the threshold) in these two

populations (*f7* and *f15*). From these two populations, two random samples were extracted independently with $n_1$ and

$n_2$ sizes. These samples had $p_1$ and $p_2$ success probability, which are constant in each extraction. Positive events

occur when the observation is equal or below the threshold.

3.  Contrast statistics:

Sample *f7*: $n_1$, $P_1$; where $n_1$ = population of *f7* and $P_1$ = ratio of positive events

Sample *f15*: $n_2$, $P_2$; where $n_2$ = population of *f15* and $P_2$ = ratio of positive events


$$P = \frac{n_1 P_1 + n_2 P_2}{n_1 + n_2}$$         (2)


$$Z = \frac{P_1 - P_2}{\sqrt{P(1-P)\left(\frac{1}{n_1} + \frac{1}{n_2}\right)}}$$         (3)


4.  Critical ratio





Bilateral:
$$Z \leq {}_{\alpha/2}Z$$
$$Z \geq {}_{1-\alpha/2}Z$$
5.   Decision.- Reject $H_0$ if contrast statistics falls in critical rate or $p \leq \alpha$
**2.7    Insurance contract design**
**2.7.1    Indemnity calculation**
The indemnity is the amount of money that an insured individual receives when a covered hazard occurs. In this case, we have
two insurance policies options. The first one is the working capital, where the insured amount corresponds to the money
necessary for recovering the investment (production cost) that a farmer has spent. The second one is the profit (gross margin),
where the insured amount is the money that a farmer would obtain selling his production after covering his production cost in
a normal year.
In other words, for the first option the compensation will cover the yield reduction between the economic and physiologic
threshold. In the second case, the compensation will cover the difference between the expected yield in a normal year and the
yield obtained at the economic threshold.
Thus, the indemnity calculation follows the next equation (Maestro et al., 2016):
$I_{sz} = Y_z \times P - Pc_{sz}$                                                      (4)
Where:
$I_{sz}$ is the net income expected per hectare (USD/ha) in a normal year, differentiated by *s* scenario (it could be 1 or 2), and *z*
zone (*f7, f15* or *cantonal*).
$Y_z$ is the expected yield (tones/ha), in normal years for *z* zone.
*P* is the price of a ton of rice at farm (USD/t).
$Pc_{sz}$ is the production cost per hectare of rice cultivation (USD/ha) , differentiated by *s* scenario (it could be 1 or 2), and *z*
zone (*f7, f15* or *cantonal*).
$Y_z$ is obtained applying Eq. (1), *P* was calculated from rice price monthly variation along the last two years. This value is
assumed to be constant (371 USD/t) for both AHZs and *cantonal*, and for scenario 1 or 2.
To estimate $Pc_{sz}$, we evaluated two scenarios. Scenario 1, with production costs differentiated for each *z* zone (*f7, f15* or
*cantonal*); and scenario 2, with the same production costs for all *z* zones (*f7, f15* or *cantonal*).
**2.7.2    Premium determination**
The commercial or loaded premium cost $CP_{sz}$ is equal to the net premium multiplied by a factor that covers the insurance
company profit and loading cost. The net premium or risk premium $NP_{sz}$ has to cover the expected compensations that an
insurance company would have to pay during the analysed period. The net premium is calculated as a percentage of $I_{sz}$. This
percentage corresponds to the probability that the insurance company have to compensate $I_{sz}$ in a period of time (Jasiulewicz,
2001; van de Ven et al., 2000). It was expected that the probability of occurrence is different for each AHZ (*f7* and *f15*). It is





also different when the NDVI_ave measure is made at *cantonal level*. Thus, we calculated differentiated premium rates for
each one of these cases.
$$NP_{sz} = I_{sz} \times Pr_{sz} \qquad (6)$$
$$CP_{sz} = NP_{sz}(1 + (\beta_1 + \beta_2)) \qquad (7)$$
Where:
$NP_{sz}$ is net premium rate (USD/ha) for scenario *s* (scenario 1 or 2) and *z* zone (*f7*, *f15* and *cantonal*).
$CP_{sz}$ is commercial premium rate (USD/ha) for scenario *s* (scenario 1 or 2) and *z* zone (*f7*, *f15* and *cantonal*).
$Pr_{sz}$ is the probability of sinister occurrence for *s* scenario (scenario 1 or 2) and *z* zone (*f7*, *f15* and *cantonal*).
$\beta_1$ is the insurance company profit (20% of $NP_{sz}$).
$\beta_2$ is the operative cost of the insurance plus taxes (5% of $NP_{sz}$).
The commercial premium value $CP_{sz}$ in index-based insurance is generally subsidized by the government in around 60% to
small farmers in developing countries (Peter Höppe, 2007; Ricome et al., 2017).
**3    Results and discussion**
**3.1    Statistical analysis**
From descriptive statistical analysis, the kurtosis (0.56) and a skewness (-0.78) indicated that the data set of NDVI_ave fits a
normal distribution; however, the Lilliefors (Kolmogorov-Smirnov) normality test showed: D = 0.080207 and a p-value < 2.2e-
[16] lower than 0.05; then we rejected the null hypothesis because the data set does not come from a normal distribution. We
have found that our data fits a Generalized-minimum extreme value (GEVmin) distribution (Kotz and Nadarajah, 2000) for
the *cantonal* data set and for the two AHZs (*f7* and *f15*) based on $\chi^2$ statistics (Table 3).
**Table 3. Parameters of Generalized-minimum extreme value (GEVmin) distribution for each AHZ and cantonal and distribution**
**adjustment statistic of maximum likelihood**

| | Mode | Scale | n | k | F.D. (n-1)x(k-1) | Chi-squared table | Chi-squared calculated |
|---|---|---|---|---|---|---|---|
| *Cantonal* | 0.52 | 0.09 | 100 | 11 | 990 | 1064.31 | 9.42 |
| *f7* | 0.51 | 0.11 | 100 | 11 | 990 | 1064.31 | 2.75 |
| *f15* | 0.53 | 0.08 | 100 | 10 | 891 | 961.55 | 2.16 |

Because the data sets did not fit normal distributions, we used a non-parametric test to determine if NDVI_ave medians in
zones *f7* and *f15* are significantly different. The Kruskal-Wallis test for these zones ($\chi^2$= 345.48, F.D. = 1, p-value < 2.2e-16)
shows us that the null hypothesis of *f7* and *f15* being equal can be rejected because the p-value is lower than 0.05. The same
test mentioned before shows us that years are also significant different ($\chi^2$= 7507.4, F.D. = 16, p-value < 2.2e-16 is also lower
than 0.05). Five categories in years are establish when LSD (Mean) and Bonferroni (Median) test are applied on NDVI_ave
values (see Table 4).





**Table 4. Fisher`s Least Significant Difference (LSD) test for comparing means, and Bonferroni test for comparing medians, for**
**years**

| Year | Mean (NDVI_ave) | Year | Median (NDVI_ave) | Category |
|------|------|------|------|------|
| 2008 | 0.39 | 2008 | 0.39 | *Very low* (Years affected by extreme climatic events) |
| 2012 | 0.40 | 2013 | 0.42 | |
| 2013 | 0.40 | 2016 | 0.43 | |
| 2016 | 0.40 | 2012 | 0.43 | |
| 2017 | 0.42 | 2017 | 0.44 | *Low* (Years affected by moderate climatic events) |
| 2014 | 0.42 | 2014 | 0.45 | |
| 2015 | 0.45 | 2010 | 0.47 | |
| 2010 | 0.46 | 2015 | 0.48 | |
| 2002 | 0.48 | 2011 | 0.48 | *Normal* (Normal years) |
| 2005 | 0.49 | 2005 | 0.49 | |
| 2011 | 0.49 | 2002 | 0.49 | |
| 2001 | 0.51 | 2001 | 0.52 | *High* (Years with good climatic conditions) |
| 2009 | 0.51 | 2009 | 0.52 | |
| 2007 | 0.52 | 2007 | 0.52 | |
| 2003 | 0.54 | 2004 | 0.54 | *Very High* (Years with very good climatic conditions) |
| 2004 | 0.55 | 2003 | 0.55 | |
| 2006 | 0.55 | 2006 | 0.56 | |

**3.2    Rice yield estimation**
The observed rice yield was plotted versus NDVI_ave in a rice crop cycle. A normal accumulative curve was adjusted, see Eq.
(1) in Fig. 3 A, to relate both variables; where μ (0.49) is the mean of NDVI_ave measured in yield sampling points of years
2014 through 2017; σ (0.14) is the standard deviation, and X is the NDVI_ave value for which, we want to estimate rice yield
(Y). The RMSE (%) of 3.3 and an $R^2$ of 0.71 indicate a robust model. This type of curve was selected, instead of a linear
regression, to take into account the high values of the NDVI saturation effect on plant biomass (Gu et al., 2013) and the soil
saturation effect on low NDVI values (Rondeaux et al., 1996). The correlation coefficient of observed versus estimated rice-
yield was high (0.89), showing that NDVI_ave is an adequate indicator for assessing the impact of drought and flood over rice
crop (Fig. 3 B).

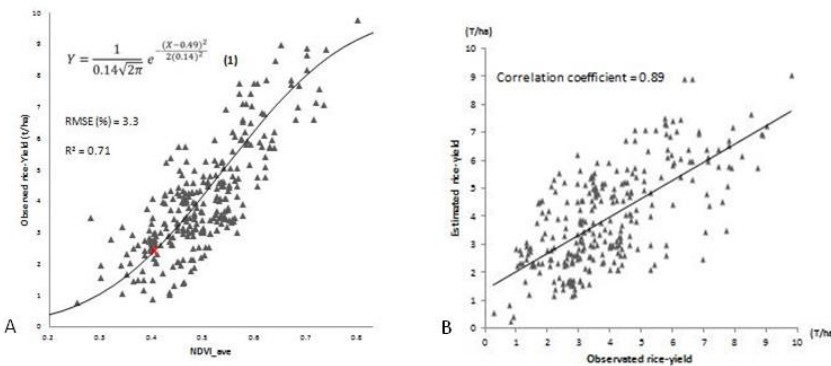


**Figure 3: (A) Scatter plot of observed rice-yield and NDVI_ave, curve of Eq. (1) for estimating yield (Valverde-Arias et al., 2019);**
**and (B) correlation of observed and estimated rice-yield**





## 3.3 Thresholds determination

Once that the years have been classified in five categories, we could define the different levels of impact or no impact over rice crop (NDVI_ave), as shown in Table 4. When rice yield is less than 0.5 t/ha (NDVI_ave ≤ 0.26), due to damage in rice crop by extreme events, the total loss threshold is neither detectable at cantonal level nor at AHZs (*f7* and *f15*) level. Individual NDVI_ave observations equal or under the total losses' threshold can be found but not as a regional measure of NDVI_ave. However, in our IBI design the index measure is an average of all observations within a homogeneous zone, being these *cantonal* or AHZs (*f7* and *f15*).

The physiologic threshold represents the maximum rice-crop damage that can be detected through NDVI_ave at regional scale, which has been caused by an extreme climatic event. It is fixed (0.4) for both AHZ (*f7* and *f15*) and cantonal (see Table 5). The years when we reached the physiologic threshold in our data set were 2008, 2012, 2013, and 2016. These years belong to the "very low category"; and according to climatic application (NOAA, 2018) these years were affected by extreme climatic events. This application contains and plots historical climatic data. In this case, we analysed the combined precipitation anomalies. Zero represents no precipitation anomaly, i.e. average precipitation; positive anomalies occur in years that precipitation is above the average (floods) and negative anomalies when the precipitation is below the average (drought). As we can see in Fig. 4 A and B, Babahoyo canton presented positive anomalies of precipitation (floods) in 2008 and 2012 and negative anomalies of precipitation (drought) for 2013 and 2016 (Fig. 4 C and D).

**Table 5. Different categories of impact thresholds for scenarios of differentiated and non-differentiated production costs**

| | Differentiated production cost (Scenario 1) | | | | | | Non differentiated production cost (Scenario 2) | | | | | | Type of Insurance | | |
| | Cantonal | | f7 | | f15 | | Cantonal | | f7 | | f15 | | Occurrence verification by | IBI | Conventional |
| | NDVI _ave | Yield (t/ha) | NDVI _ave | Yield (t/ha) | NDVI _ave | Yield (t/ha) | NDVI _ave | Yield (t/ha) | NDVI _ave | Yield (t/ha) | NDVI_ ave | Yield (t/ha) | | Compensation for | |
|---|---|---|---|---|---|---|---|---|---|---|---|---|---|---|---|
| **Expected Yield** | 0.51 | 5.65 | 0.49 | 5.11 | 0.55 | 6.68 | 0.51 | 5.65 | 0.49 | 5.11 | 0.55 | 6.68 | Index | NSC | NSC |
| **Economic Threshold** | ≤0.43 | 3.39 | ≤0.41 | 2.75 | ≤0.47 | 4.39 | ≤0.43 | 3.39 | ≤0.43 | 3.39 | ≤0.43 | 3.39 | **Index** | **Profit** | **NSC** |
| **Physiological Threshold** | ≤0.40 | 2.65 | ≤0.40 | 2.65 | ≤0.40 | 2.65 | ≤0.40 | 2.65 | ≤0.40 | 2.65 | ≤0.40 | 2.65 | Index/in-situ | Profit | Investment |
| **Total losses** | ≤0.26 | ≤0.5 | ≤0.26 | ≤0.5 | ≤0.26 | ≤0.5 | ≤0.26 | ≤0.5 | ≤0.26 | ≤0.5 | ≤0.26 | ≤0.5 | In-situ | No detectable | Investment |

IBI (index based Insurance); NSC (not susceptible of compensation)

On the other hand, the economic threshold depends on economic factors such as sale rice price and production cost. These are not constant, and must be set regarding the necessary yield for covering the farmer's expenses during the rice cultivation campaign, as it is shown in Table 5.

The economic threshold represents the minimum yield that farmers must reach for covering at least the production cost. It is higher than physiologic threshold, and it varies according to the scenario. In scenario 1, the economic threshold is different for each AHZ (*f7* and *f15*); *f7*'s being lower production cost (1022 USD/ha) than that for *f15* (1629 USD/ha). Thus, the economic threshold of *f7* is 0.41, while for *f15* is 0.47 (see Table 5). The years from our dataset that reached the economic threshold were 2010, 2014, 2015 and 2017. They were impacted by moderate climatic events (flood for 2010 and 2017 and drought for 2014 and 2015) according to NOAA (2018), see Fig. 4 E, F, G and H.

For scenario 2, the production cost is a weighted average (1259 USD/ha) both for AHZs (*f7* and *f15*) and cantonal. Therefore, the economic threshold (0.43) is the same for AHZs and *cantonal*, see Table 5.




Figure 4: Positive anomalies of precipitation (flood) in (A) year 2008, (B) year 2012, (E) year 2010, (F) year 2017; negative
anomalies of precipitation (drought) in (C) year 2013 and (D) year 2016, (G) year 2014 and (H) year 2015. Source: (NOAA, 2018)




**3.4    Risk assessment of AHZ and Babahoyo canton**
The risk status of *f7* and *f15* were found to differ based on the following results. 25% of events were found under the physiologic
threshold for *f7* and 17% for *f15* (see Fig. 5 B, C); and when we do not consider AHZs (*cantonal*) 21% (Fig. 5A). AHZ *f7*'s
probability is higher because of its soil conditions (see Table 6). These conditions make the zone more vulnerable to floods
due to its very fine texture (>60% clay), flat lands (0-5% slope), very low altitude (1-12m) and proximity with rivers' banks
that contribute to very poor drainage of this zone. In the same way, these characteristics could give to *f7* better capacity for
long-term water retaining, during a drought. However, when drought is extreme, the *f7*'s soil (Vertisol) gets very dried (Soil
Survey Staff, 2014); consequently, it becomes very hard and develops deep cracks. This phenomenon affects physically the
crop's roots and hinders considerably the soil tillage (Valverde-Arias et al., 2019).

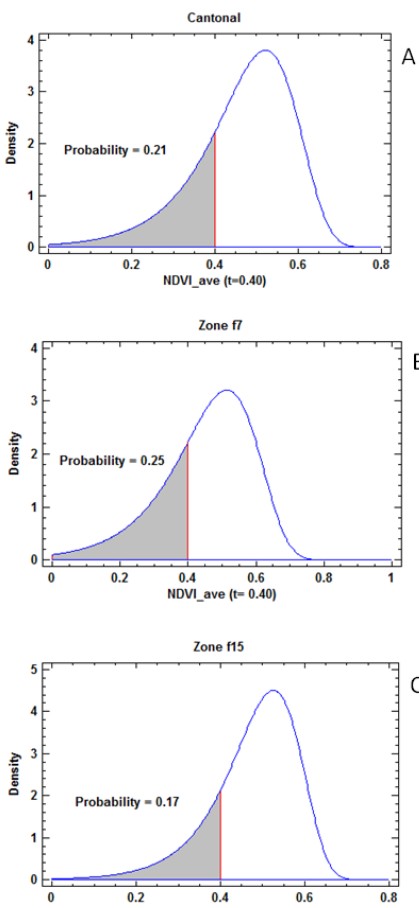


**Figure 5: Physiologic threshold (red line) within Generalized-minimum extreme value (GEVmin) distribution of NDVI_ave; (A)**
***cantonal, (B) f7 and (C) f15 zones***






**Table 6. Soil and climatic characteristics of Agro-ecological homogeneous zones AHZs in Babahoyo Canton**

|  | Zone *f7* | Zone *f15* |
|---|---|---|
| Slope | 0-5% | 5-12% |
| Altitude | 1-12m | >12-35 |
| Clay | >50% | 35-50% |
| Effective depth | 50-100 cm | >100 cm |
| pH | 5.6-6.5 | 6.6-7.4 |
| Organic matter | 2-4% | 2-4% |
| Temperature | 24-25 ℃ | 24-25 ℃ |
| Precipitation | 500-700 mm | 700-900 mm |
| Soil Classification* | Typic Hapluderts | Vertic Eutrudepts |

Source: (Valverde-Arias et al., 2019); *According to USDA Soil Taxonomy (Soil Survey Staff, 2014)
For economic thresholds, we also found differences between the risk status of AHZs *f7* and *f15*. Furthermore, for scenario 1,
the probability of having events equal or under the economic threshold is higher in *f15* (37%) than that in *f7* (26%) and that in
*cantonal* (29%), as we can see in Fig. 6 A, C and E. The reason for this is that in this scenario, *f15*'s farmers have to cover a
higher production cost (which corresponds to semi-technical production system), and, therefore, they have to reach an
economic threshold also higher (0.47) than that one in the *f7*.

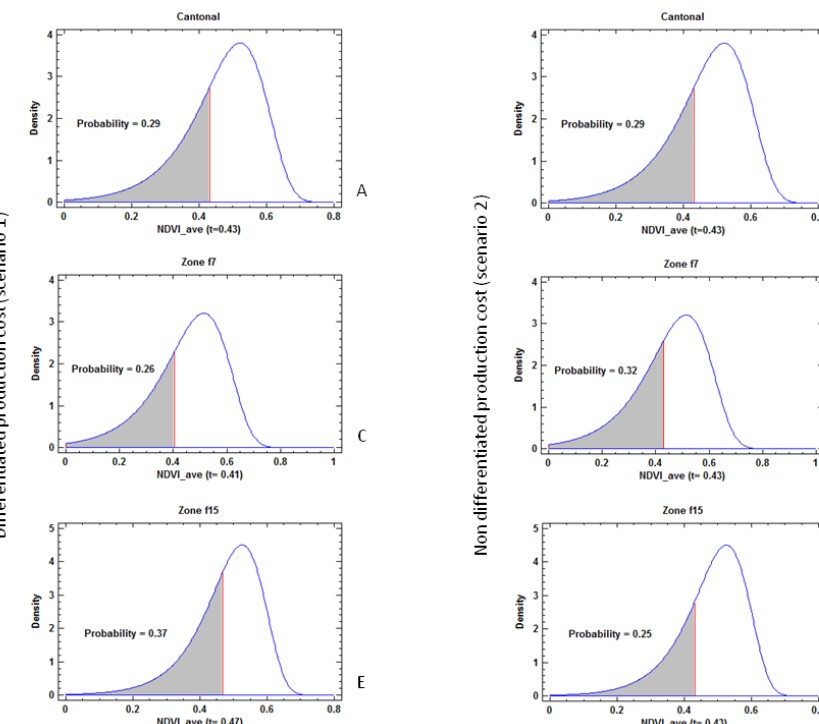


**Figure 6: Economic threshold (red line) within Generalized-minimum extreme value (GEVmin) distribution of NDVI_ave for**
**differentiated production cost (scenario 1) for: (A) *cantonal*, and Agro-ecological homogeneous zones (C) *f7* and (E) *f15*; and for**
**non-differentiated production cost (scenario 2) for: (B) *cantonal*, and Agro-ecological homogeneous zones (D) *f7* and (F) *f15***




In scenario 2, the economic threshold is equal (0.43) for *f7*, *f15* and *cantonal*, but the probability to find events under the
threshold is higher in *f7* (32%) than that in *f15* (25%) and that in *cantonal* (29%). Although the economic threshold is the same
for both AHZs (*f7* and *f15*) and at *cantonal* level, in this scenario, the frequency distributions of NDVI_ave were different for
each zone. Consequently, they accumulated different probabilities under the same threshold, as shown in Fig. 6 B, D and F.
At this point, we evaluated the Z-test results for determining if the found differences have statistical significance. Based on
the Z-test (see Table 7), the null hypothesis ($H_0: p_1 = p_2$) can be rejected in both scenarios 1 and 2, so we can assert that the
proportion of positive cases (equal or under physiologic and economic thresholds) in *f7* are significantly different from that in
*f15*. For economic threshold in scenario 1 (differentiated production cost), the calculated Z is negative because in this case the
probability in *f15* is higher than in *f7*.
**Table 7. Z-test for probability of events susceptible of compensations ESC under physiologic and economic thresholds in Agro-**
**ecological homogeneous zones (*f7* and *f15*) and cantonal**

|  | Type of threshold | Observations | Positive events | Probability | Critical rate | Z test |
|---|---|---|---|---|---|---|
| Non differentiated production cost/ differentiated production cost | *Physiological threshold cantonal* | 31,756 | 6669 | 0.21 | Z≤ z_0.025= -1.96 Z≥z_0.975=1.96 | **18.06** |
|  | *Physiological threshold f7* | 13,498 | 3375 | 0.25 |  |  |
|  | *Physiological threshold f15* | 18,258 | 3041 | 0.17 |  |  |
| Differentiated production cost (scenario 1) | *Economic threshold cantonal* | 31,756 | 9209 | 0.29 | Z≤ z_0.025= -1.96 Z≥z_0.975=1.96 | **13.59** |
|  | *Economic threshold f7* | 13,498 | 4320 | 0.32 |  |  |
|  | *Economic threshold f15* | 18,258 | 4565 | 0.25 |  |  |
| Non differentiated production cost (scenario 2) | *Economic threshold cantonal* | 31,756 | 9209 | 0.29 | Z≤ z_0.025= -1.96 Z≥z_0.975=1.96 | **-21.35** |
|  | *Economic threshold f7* | 13,498 | 3510 | 0.26 |  |  |
|  | *Economic threshold f15* | 18,258 | 6756 | 0.37 |  |  |


As NDVI_ave dataset fits a GEVmin distribution, we used this distribution, with its specific parameters (Mode and Scale,
shown in Table 3) for estimating NDVI_ave density frequencies for *f7*, *f15*, and *cantonal*. With these distributions, we
calculated the positive events under physiological and economic thresholds in scenarios 1 and 2. Then, we estimated the basis
risk of these calculations. In this case, basis risk could arise if the estimated distribution does not fit properly with the
distribution observed from measured data.
We have found that the basis risk for this estimation is negligible according to Adjusted R-squared shown in Table 8. Therefore,
we can confidently use these estimations for determining the events proportion that reached the physiologic and economic
thresholds, i.e. the occurrence probability of extreme events that warrant compensations.







**Table 8. Simulation of NDVI_ave by GEVmin distribution for AHZs and cantonal, and basis risk calculation between observed**
**(O) and estimated (E) frequencies of NDVI_ave**

| NDVI_ave Class | Frequency | | | | | |
|---|---|---|---|---|---|---|
| | O f7 | E f7 | O f15 | E f15 | O Cantonal | E cantonal |
| 0.08 | 25 | 204 | 8 | 73 | 330 | 34 |
| 0.16 | 310 | 391 | 171 | 117 | 434 | 482 |
| 0.24 | 758 | 590 | 462 | 357 | 957 | 1228 |
| 0.32 | 1215 | 1098 | 817 | 869 | 2032 | 2036 |
| 0.40 | 1963 | 1912 | 1583 | 2125 | 4119 | 3569 |
| 0.48 | 2929 | 2902 | 4751 | 4417 | 7345 | 7705 |
| 0.56 | 3413 | 3356 | 6863 | 6300 | 9375 | 10,267 |
| 0.64 | 2206 | 2355 | 3232 | 3690 | 6141 | 5404 |
| 0.72 | 598 | 653 | 354 | 310 | 1004 | 933 |
| 0.80 | 78 | 37 | 17 | 0 | 19 | 95 |
| 0.88 | 3 | 0 | 0 | 0 | 0 | 3 |
| Total Observations | 13,498 | 13,498 | 18,258 | 18,258 | 31,756 | 31,756 |
| Adjusted R-Squared | $R^2_{f7}$ = 0.99 | | f15 = 0.98 | | Cantonal = 0.98 | |


## 3.5 Indemnity calculation

The indemnity for farms that reach the physiological threshold in scenario 1 is reported in Table 9. These values show us the
deficit (negative numbers) that farmers face for recovering their production costs when their crop yield falls below the break
even, in each AHZ (*f7* and *f15*) and *cantonal*. The indemnity would make up the difference between crop's costs and revenue
in case of extreme event. In *f7* the indemnity would be 38 USD/ha, it means that when a farmer reaches physiologic threshold,
he only lacks 38 USD/ha for covering his production cost. A farmer from this scenario could dispense with the insurance
contract, because the deficit to hit the break-even is not representative. On the contrary, when *f15* reaches the physiologic
threshold, its deficit is very high (645 USD/ha), which is the money that an *f15*'s policyholder would receive as compensation
in case of an extreme event occurrence.
**Table 9. Indemnity calculation for physiologic threshold in Agro-ecological homogeneous zones (*f7* and *f15*) and *cantonal***

| | Expected yield at physiologic threshold (tonnes/ha) | Price (USD/ton) | Gross incomes (USD/ha) | Differentiated production cost (USD/ha) | Deficit for covering Investment (USD/ha) |
|---|---|---|---|---|---|
| Cantonal | 2.65 | 371.5 | 984.4 | 1259.00 | **-274.6** |
| f7 | 2.65 | 371.5 | 984.4 | 1022.00 | **-37.6** |
| f15 | 2.65 | 371.5 | 984.4 | 1629.00 | **-644.6** |


For scenario 2 of physiological threshold, the indemnity would be 275 USD/ha for all AHZ (*f7* and *f15*) and for *cantonal*, due
to their same production cost (1259 USD/ha). As was mentioned before, in Ecuador, it already exists an agricultural
conventional insurance that covers the rice growers' working capital; but we included this calculation as an alternative to
conventional insurance or for these areas where conventional insurance is not feasible.
When looking at the economic threshold, as we can observe in Table 10, the indemnity (Gross Margin) in scenario 1 is very
similar between AHZs (*f7* and *f15*) and *cantonal* though their expected yields are different. This is because their assigned




production cost has been related with their expected yield. For example, since farmers have invested more money in their crop
in *f15*, their expected yield is higher. Moreover, the difference in the premium price of these zones will be determined by the
different probability of extreme events occurrence in each AHZ (*f7* and *f15*) and *cantonal*.
**Table 10. Indemnity calculation (gross margin) for economic threshold, for each AHZ (*f7* and *f15*) and cantonal, both in scenario 1**
**and 2**

|  | Expected Yield (t/ha) | Price (USD/t) | Gross incomes (USD/ha) | Production cost scenario 1 (USD/ha)* | Gross margin scenario 1 (USD/ha) | Production cost scenario 2 (USD/ha) ** | Gross margin scenario 2 (USD/ha) |
|---|---|---|---|---|---|---|---|
| *Cantonal* | 5.65 | 371 | 2099 | 1259 | **840** | 1259 | **840** |
| *f7* | 5.11 | 371 | 1899 | 1022 | **877** | 1259 | **640** |
| *f15* | 6.68 | 371 | 2482 | 1629 | **853** | 1259 | **1223** |

* = Differentiated production cost and ** = Non differentiated production cost
In scenario 2, on the other hand, we have assumed the same production cost for *f7* and *f15*; thus, *f15* has higher expected yield
in normal years than *f7*. Obviously, in this scenario *f15* obtains the highest gross margin (1223 USD/ha), having also the highest
compensation, which would be reflected in a higher premium cost. However, *f7* has the lowest insured amount (640 USD/ha),
so that its premium cost should be low. But we have to consider that premium cost calculation also depends on the occurrence
probability of the insured event.
For economic threshold, the indemnity calculation (840 USD/ha) for *cantonal* is equal in both scenarios 1 and 2; as shown in
Table 10. Because, we used the same weighted average as production cost (1259 USD/ha). For *f7* it is expected a higher gross
margin in scenario 1 than that in scenario 2, due to scenario 2's production cost being higher. On the contrary, for *f15* the gross
margin is higher in scenario 2 than that in scenario 1; because in scenario 2, *f15* has lower production cost than in scenario 1.
**3.6  Premium determination**
The premium value is related to the insured amount (the indemnity or compensation that insurance company must pay to
farmers when an insured extreme event occurs), and the probability of the ensured extreme event occurs in a determined period.
Table 11 shows the net and commercial premium calculation for the two different thresholds under both scenario1 and scenario
2, and for each AHZ and at *cantonal* level.
In general terms, it can be appreciated that premium cost for economic thresholds are more expensive than that for physiologic
threshold, in both scenarios (1 and 2). This is because the insured amounts for economic threshold are higher than that for
physiologic threshold. In the first case, the compensation covers the entire lost profit; while in the second one, the compensation
covers only the deficit necessary for recovering the investment (production cost).
If the insured amounts are similar among AHZ (*f7* and *f15*) and *cantonal*, the difference in premium cost is determined by the
occurrence probability. However, when there are sharp differences among insured amounts of AHZs (*f7* and *f15*) and *cantonal*,
these are more determinant in the premium cost variation than the occurrence probability.
Moreover, for physiologic threshold in scenario 1, the premium cost is determined mainly by the insured amount, for instance,
for *f15* the premium cost is the highest (136.98 USD/ha) despite of its occurrence probability being the lowest. On the contrary,
for *f7* its premium cost is very low, despite its highest occurrence probability, because of having a greater insured amount.
While under economic threshold in scenario 1, the insured amount of AHZs (*f7* and *f15*) and *cantonal* are similar, the premium
cost for *f15* is the highest (394.66 USD/ha), due to its highest occurrence probability.





When costs are not differentiated across AHZ (scenario 2), for the physiologic threshold the insured amount is equal in all
AHZs (*f7* and *f15*) and *cantonal*, and thus their premium cost has been differentiated through the occurrence probability, being
the highest for *f7* (85.82 USD/ha). In the same scenario, for the economic threshold *f15* has the highest gross margin, and
therefore a high-insured amount despite its low occurrence probability (0.25). It has a high premium price (382.29 USD/ha),
but it is lower than in scenario 1 (394.66 USD/ha) where the occurrence probability is the highest (0.37).
As it can be appreciated in Table 11, after we divided the study area through AHZs map in *f7* and *f15* zones, we can perform
more accurate calculations and reduce basis risk of the premium costs according to the expected yield, insured amount, and
occurrence probability of each AHZ (*f7* and *f15*). This means that by differentiating the study area through AHZs, we can
design an accurate insurance policy where farmers from each zone pay a premium that corresponds to the risk that they are
facing. To illustrate this, for physiological threshold in scenario 1, if we do not divide Babahoyo canton through AHZs and
instead use *cantonal* as IIA, an average Babahoyo's producer (≈20 ha) from *f15* would pay only 72.09 USD/ha as insurance
premium. But if an extreme event occurs, he would receive as compensation only 4915.68 USD which is less than half of the
actual loss in a year that an extreme event occurs (11,796.56 USD). At the same time, for the same threshold and scenario,
farms from *f7* would pay a much lower premium (11.76 USD/ha), and in case of disaster receive a small compensation which
is adjusted to the actual losses experienced by the farmers. This can be of great relevance, as if we assume that farms in *f7* are
non-technical production systems that achieve lower yields and get lower economic returns, providing access to affordable
insurance with fair premium prices may importantly contribute to expand insurance uptake and reduce substantially socio-
economic vulnerability in this area.
**Table 11. Calculation of commercial premium rate for physiologic and economic thresholds in scenarios 1 and 2 and for AHZ (*f7***
**and *f15*) and *cantonal***

| Threshold type | Zone | Threshold value | Insured amount (USD/ha) | Occurrence probability of IEE | Net premium cost (USD/ha) | Commercial premium cost (USD/ha) | Production cost + subsidized premium cost (USD/ha) | Compensation to a policy holder of a farm of 20 ha (USD)* |
|---|---|---|---|---|---|---|---|---|
| *Scenario 1 (Differentiated production cost)* | | | | | | | | |
| Physiologic | *Cantonal* | 0.40 | 274.62 | 0.21 | 57.67 | **72.09** | 1287.83 | 4915.68 |
| | *f7* | 0.40 | 37.62 | 0.25 | 9.4 | **11.76** | 1026.70 | 658.32 |
| | *f15* | 0.40 | 644.62 | 0.17 | 109.59 | **136.98** | 1683.79 | 11,796.56 |
| Economic | *Cantonal* | 0.43 | 840.21 | 0.29 | 243.66 | **304.58** | 1380.83 | 14,367.56 |
| | *f7* | 0.41 | 877.28 | 0.26 | 228.09 | **285.12** | 1136.05 | 15,264.64 |
| | *f15* | 0.47 | 853.31 | 0.37 | 315.73 | **394.66** | 1786.86 | 13,908.92 |
| *Scenario 2 (Non differentiated production cost)* | | | | | | | | |
| Physiologic | *Cantonal* | 0.40 | 274.62 | 0.21 | 57.67 | **72.09** | 1287.83 | 4915.68 |
| | *f7* | 0.40 | 274.62 | 0.25 | 68.65 | **85.82** | 1293.33 | 4805.84 |
| | *f15* | 0.40 | 274.62 | 0.17 | 46.68 | **58.36** | 1282.34 | 5025.52 |
| Economic | *Cantonal* | 0.43 | 840.21 | 0.29 | 243.66 | **304.58** | 1380.83 | 14,367.56 |
| | *f7* | 0.43 | 640.28 | 0.32 | 204.89 | **256.11** | 1361.44 | 10,756.72 |
| | *f15* | 0.43 | 1223.32 | 0.25 | 305.83 | **382.29** | 1411.91 | 21,408.08 |

* In a year when an ensured extreme event (drought and flood) occurs, 20 ha is the average size of a rice-farm in Ecuador
Yet, the price of the premium could be expensive for some farmers, but we must consider that this insurance will cover both
of the most frequent and intense extreme events that affect Babahoyo canton (drought and flood). For example, for the
economic threshold in scenario 1, the premium cost without subsidy would reach the 22% of the total production cost of a
policy holder of *f7* and the 20% for *f15*. This means that subsidizing premium cost may still be necessary in order to incentivize



the insurance contract taking (Garrido and Zilberman, 2008; Yuanchang and Jiyu, 2010), and the Government subsidy of 60%
of the premium cost that it is currently offered in Ecuador with the conventional insurance would still be required.
Furthermore, if Government would apply prevention policies to promote farms' modernization, farmer's technical training,
and civil works the occurrence probability of extreme events could be reduced or at least mitigated. For instance, dams and
irrigation infrastructure could improve the risk status of Babahoyo's farmers facing drought and floods. Consequently, it could
be reflected in an insurance premium price reduction.

## 4   Conclusions

Floods and droughts are a major threat for rice production in Ecuador that undermine food security and endanger sustainability
of rural livelihoods in many areas of the country. Risk management mechanisms, such as agricultural insurance, may play an
important role in stabilizing production and contributing to reduce the vulnerability of rice farmers. In this context, IBI is a
promising tool that facilitates the implementation of agricultural insurance and reduces operational and transaction costs.
However, basis risk may lead to inadequate premium prices and to unfair indemnity calculations and payment. To avoid this,
the identification of an adequate index and a proper knowledge of variability within the IIA are crucial.
In this research, we developed an IBI based on NDVI_ave that accounts for variability across the insured area. For this, we
considered AHZs as the starting point for risk assessment and indemnity calculation and compared it with the insurance design
at cantonal level. Two levels of climatic impact over rice cultivation have been identified. The first one is the physiological
impact that is determined by a physiological threshold when a climatic event is extreme, its policy contract will cover losses
related to the rice grower's working capital. The second level is the economic impact when the climatic event is moderate, and
its policy will cover the crops' gross margin.
The results of the analysis performed evidence that the two AHZs show significantly different risk profiles for physiologic and
economic thresholds. Therefore, the design of differentiated premium calculation based on the risk status and insured amount
of each AHZ ($f7$ and $f15$) will facilitate that farmers pay a fair insurance premium. This insurance premium would be as
consistent as possible with their risk status and would help them to receive compensations that effectively cover the totality of
their losses.
The basis risk arising from modelling the risk frequency of drought and flood events in Babahoyo (cantonal) and in AHZs ($f7$
and $f15$) through GEVmin distribution is negligible. The basis risk associated with the spatial heterogeneity of Babahoyo
canton has been reduced in our IBI design. We have accomplished this by dividing this canton into $f7$ and $f15$ homogeneous
zones which have a significant different risk status, different expected yields and may have also different production costs.
Regarding all these factors and the two different impact levels for the IBI design, have allowed to set up a fair premium and
reduce in this way the possible bias caused for not discriminating Babahoyo variability.
The cost for contracting an insurance policy could be expensive in some cases. However, the fact that this kind of insurance is
generally partially subsidized by the government in developing countries (as Ecuador) could make this insurance affordable
to farmers. Moreover, even if the premium price may be high, the index design guarantees to policyholders that the premium
price is fair and proportional with the risk they are facing.
The implementation of IBI for rice crop in Babahoyo could let Ecuadorian Government to respond efficiently and rapidly in
the case of an extreme climatic event, paying compensations faster than with the conventional insurance. It could stabilize
rice-producer incomes and reduce small farmers' vulnerability by providing access to insurance through premium and


indemnities adjusted to the specific risk and technology conditions. Consequently, it can incentivise rice cultivation to the
desirable levels for covering national demand ensuring food security of Ecuador.
Finally, it is worth mentioning that even if the IBI has been defined for rice crop in a particular area, the methodology applied
for developing such an insurance scheme can be applied for other crops and regions if the data to define AHZs, NDVI
distributions, crop yield and cost productions are available. This is, therefore, a promising approach for defining IBI schemes
minimizing basis risk, which can importantly profit from current advances in remote sensing, satellite imagery and improved
information systems.

**Author contribution**

Omar Valverde has developed the research idea and write the original draft of the manuscript, guided and supervised by
Alberto Garrido, Ana Tarquis and Paloma Esteve. Ana Tarquis has contributed with the imagery and statistical analysis and
generation of agro-ecological homogeneous zones. Alberto Garrido contributed with the insurance design and Paloma with
the policy and socio-economic implications of the insurance implementation. Alberto, Ana and Paloma have reviewed and
edited the manuscript for obtaining the final version.

**Competing interest**

The authors declare that they have no conflict of interest

**Acknowledgements**

Financial support for Omar Valverde Ph.D. studies in UPM by National Secretary of Higher Education Science Technology
and Innovation of Ecuador (SENESCYT in Spanish) is greatly appreciated. The authors would like to also acknowledge to
CGSIN-MAG for the data provided. The funding from by the Comunidad de Madrid (Spain) and Structural Funds 2014-2020
(ERDF and ESF), through project AGRISOST-CM S2018/BAA-4330, are highly appreciated.

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
