# Peer review of "Remote sensing in an index-based insurance design for hedging economic impacts on rice cultivation"

_Natural Hazards and Earth System Sciences, 2019_

## Referee Comment (RC1) · Anonymous Referee #1 · 13 Aug 2019

The paper presents the definition of an index-based insurance for possible losses on rice production due to weather extreme events using remote sensing and field data. The paper well fits with the journal and the special issue and is recommended for publication after some possible improvements. Despite the number of comments, these are not related to substantial changes, but in some cases alternative approaches should be considered and some results should be better presented.

Comments on data, methods and results. 1) Have you considered the option of using EVI instead of the NDVI? In some cases, it can show better estimates of yeld than NDVI. It could be useful to motivate the choice. 2) Have you used quality information to

filter MODIS data considered in the analysis? Being the rainy season there can be a high influence of clouds, but it seems that quality information have not been considered. This can be explained and motivated. 3) Have you considered the possibility of using other statistics of NDVI (max, percentiles?). It can interesting to see if the average was the best one. 4) You define extreme events based only on precipitation, but this rather a big simplification. The description of what you consider extreme event should be more evident and I would suggest explain why you adopted this simplified approach. Further, since the two considered zones have rather different precipitation regimes, should extreme precipitation-related event have different thresholds? 5) Looking at Figure 3A, is seems that even a linear fitting could give a good result: have you tried it? How large is the difference with the normal accumulative curve? 6) Looking at figure 3B is possible to see several cases of large under/over production estimates: have you checked those cases?

Specific comments on figures. Figure 1: this figure can be merged with Figure 2 saving space (maybe Latin America map can be removed) and figure 1A is similar to figure 2 but not dividing coloured area in zones. Figure 2: see above Figure 3: the graph in figure A seems that could be fitted also linearly: how different would it be? Figure 4: images should all use the same legend, otherwise the comparison lead to wrong interpretation. Further, reducing the area to a smaller region can simplify making legend uniform. Figure 5: axis should be the same in all graphs. Figure 6: axis should be the same in all graphs.

Specific comments on tables. Table 1: this table do not provide many additional information and can be removed, if you want. Table 6: The use of the symbol > is a bit misleading, so I'd remove it and the unit of precipitation maybe is missing the time period. Table 7: it seems there is some inconsistency between table and text, since Z is negative in Scenario 2 and not in Scenario 1. Further, many information are repeated in the table: would it possible to remove repetition and improve the readability? Table 8: the first column is a class, but it show just a single value. Maybe it would be better to

show the boundary values of each class. The table could be also replaced by a graph. Tables 9 and 10: would it be possible to merge these tables? Further why the price should be always either 371 or 371.5. Table 11: it might be useful to add a column with the compensation per ha, before the total compensation for a 20 ha farm.

―――――――――――――――――

---

## Referee Comment (RC2) · Anonymous Referee #2 · 14 Sep 2019

I. General comments: The paper is within the scope of the journal. The paper is characterized by scientific innovation and generality, since the methodology can be applied in different regions and climatic environments. In that sense, it is extremely useful presentation of a new approach. The paper is very well and analytically presented with detailed description of the steps followed and explanation of the corresponding results.

II. Technical-scientific comments: 1. Lines 79 and 118. In reviewing the existing remote sensing literature, it is worth to mention the system Copernicus, with Sentinel 1, or 2 or 3, which is very promising with improved resolution (20-30 m) and what is the expected improvement in the accuracy of the analysis. The authors should add comments and

references, and/or comparisons. 2. Lines 73, 83-84 and 113. Agroecological zones. The proposed methodology uses principal component analysis (PCA) for classification. The subject of agroclimatic classification and zoning is large, which considers different methodologies, including combinations of satellite indices, besides PCA. The authors should mention the main existing literature in the subject, since there might be potential improvement of the conducted agroecological zoning. 3. Line 127. A clarification is required about the NDVI_ave data used: (1) these are annual values?, (2) which means that for each pixel there are 17 values? (3) How the sample points of 30% are selected? randomely? Why only 30%? (4) the rest are used for validation or not used at all? 4. Line 142. For rice estimation, what NDVI_ave values are used? annual per pixel?

III. Technical corrections: 1. Line 102. Never a sentence starts with number...: instead of 84%, write: " , where 84%...". 2. Line 120. Instead "... the precipitations", write "rainy season". 3. Line 277. Instead of "Once that...", write "Since..." 4. Line 304. Never a sentence starts with number...: instead of 25%, write: " , where 25%...".
* * *

---

## Author Comment (AC1) · 1 Nov 2019

We are grateful for your comments and questions that for sure have contributed to the improvement of this manuscript. Please see the detailed responses attached in the supplementary file and the corrected version of the manuscript

Please also note the supplement to this comment: https://www.nat-hazards-earth-syst-sci-discuss.net/nhess-2019-183/nhess-2019-183-AC1-supplement.zip

2019-183, 2019.

---

## Author Comment (AC2) · 1 Nov 2019

Thank you very much for your positive comments, questions and suggestions for improvement. Below in supplement file you can find the responses to your comments and the changes made to the manuscript.

Please also note the supplement to this comment: https://www.nat-hazards-earth-syst-sci-discuss.net/nhess-2019-183/nhess-2019-183-AC2-supplement.zip

2019-183, 2019.

---

## Author Response (AR1)

The paper presents the definition of an index-based insurance for possible losses on rice production due to weather extreme events using remote sensing and field data.

The paper well fits with the journal and the special issue and is recommended for publication after some possible improvements. Despite the number of comments, these are not related to substantial changes, but in some cases alternative approaches should be considered and some results should be better presented.

We appreciate and thank reviewer's comments and we revised the manuscript according to them.

Comments on data, methods and results.

1) Have you considered the option of using EVI instead of the NDVI? In some cases, it can show better estimates of yield than NDVI. It could be useful to motivate the choice.

Thank you for the comment and suggestion. We have tested different vegetation indexes at the beginning of this research (NDMI, NDFI, SAVI and EVI); being NDVI and EVI the ones with the best performance both detecting drought and flood impacts and estimating yield. Between NDVI and EVI, we did not find significant differences. Thus, we finally chose NDVI because this index is the most used and documented in similar studies. We think that EVI could have some advantages if we would use not highly-processed imagery where atmospheric correction could make the difference.

See in the manuscript **lines 80-84**

Particularly, the Normalized Difference Vegetation Index (NDVI) or the Enhanced Vegetation Index (EVI) are the ones performing the best in terms of detection of drought and flood impacts and estimating yields, being NDVI one of the most used in crop monitoring and current IBI systems as mentioned in many literature, e.g. (Rhee et al., 2010; Van Tricht et al., 2018; Vroege et al., 2019; Zhang et al., 2017)

2) Have you used quality information to filter MODIS data considered in the analysis? Being the rainy season there can be a high influence of clouds, but it seems that quality information have not been considered. This can be explained and motivated.

Yes, we considered information on data quality and explained it in the revised version of the paper (**see lines 145-148**). The MODIS imagery is a highly processed product in order to use the best available pixel in the composite image. Additionally, we used the Quality Assurance layer (quality layer: 250m 16 days VI Quality) included in the HDF file of NDVI MODIS imagery. In this layer, each pixel has a rank key that identifies the pixel quality, the rank key = 0 means good data, use it with confidence. Then, we used only pixels with 0 rank key following MODIS Vegetation Index User's Guide (Didan et al., 2015).

3) Have you considered the possibility of using other statistics of NDVI (max, percentiles?). It can be interesting to see if the average was the best one.

The majority of insurance cases use historical data for set a curve of NDVI mean along the year and they consider between one or two standard deviations under the mean for identifying the occurrence of an extreme event impact, as the case of Spain. They have determined that the NDVI average is an adequate crop state estimator (Dario, 2015; Vrieling et al., 2014; Vroege et al., 2019). Our approach is similar but instead of monitoring the NDVI along each stage of rice crop cycle during rainy season, we used the NDVI average of the entire rice crop cycle. If the impact of an extreme event occurs in a particular stage of rice-crop cycle, it is going to be reflected in the NDVI_ave.

4) You define extreme events based only on precipitation, but this rather a big simplification. The description of what you consider extreme event should be more evident and I would suggest explain why you adopted this simplified approach. Further, since the two considered zones have rather different precipitation regimes, should extreme precipitation-related event have different thresholds?

Actually, we defined extreme events through their impact over rice crop, which is evidenced in NDVI of rice cultivation. The climatic data is not enough robust yet in Ecuador, for using a climatic index as precipitation, temperature, evapotranspiration or hydric balance. Thus, we used a vegetation index. We have analysed the NDVI time series imagery of rice crop cycle during seventeen rainy seasons; in order to identify anomalous years. The years had been classified in five categories, according to their NDVI mean and median (LSD and Bonferroni analysis); the two lowest categories would correspond to extreme-climatic impacted years, respect to regular years. We already knew that the years from the two lowest categories were impacted by droughts and floods, because they are the only events capable to have a regional extend which impact can be detected through satellite imagery. Moreover, these years have been widely documented, because these phenomena (drought and flood) are mostly caused by El Niño Southern Oscillation (ENSO), that not only affects agriculture but infrastructure and people in general. Thus, in order to have a confidence source of climatic data, we used the NOAA climatic application, through which we have found that the years from the lowest categories also present precipitation anomalies (drought and flood). In this way, we established the relationship between the impact over rice crop (evidenced in low NDVI) and the cause of that impact, the extreme events (drought and flood).

With respect to the use of different thresholds for each zone, our thresholds are not climatic but physiologic and economic. The physiologic threshold cannot be differentiated; because it represents a crop damage threshold, the damage level is independent of the zone. It depends on the intensity of the extreme event. Thus, for reaching the same damage level in both zones, we need to have an event with the same intensity. However, the probability to have an event with a determined magnitude in each zone is different, as it is shown in the risk status analysis. We used these probabilities in our calculations. For this reason, it is not necessary to set a different physiologic threshold in each zone. On the other hand, the economic threshold was differentiated according to the conditions of each zone.

5) Looking at Figure 3A, is seems that even a linear fitting could give a good result: have you tried it? How large is the difference with the normal accumulative curve?

Yes, we tried the linear regression and it fits well, but it has a problem with NDVI values over 0.8 and bellow 0.2. It is because, NDVI is saturated and it does not respond linearly over 0.8 values, on the contrary, NDVI below 0.2 does not correspond to crop coverage. Being the normal accumulative curve the one that fits better for yield estimation.

6) Looking at figure 3B is possible to see several cases of large under/over production estimates: have you checked those cases?

Yes, we noticed that. It could be because we did not have control on the yield-sampling methodology. We had to use geo-referenced yield data from another project that fit with our study area spatially and temporary. We believe that, if we could have access to data more adequate to our working scale, we would get better results. However, despite this, the statistical analysis shows us that the majority of the sampled NDVI and yield correlated properly.

It is explained in **lines 187-193**

The General Coordination of the National Information System (CGSIN-acronym in Spanish-) of Ecuadorian Agricultural and Livestock Ministry (MAG) has conducted a rice-yield estimation project since 2014 when it began sampling yields across mapped rice areas (Moreno, 2014). Thus, 369 georeferenced rice-yield observations (t/ha) were available for 2014-2017 rainfed cycles (January to May) in the study area over AHZs *f7* and *f15* (see, Fig. 1 C). Therefore, we used these rice yield observations with their corresponding spatial and temporal NDVI_ave values for obtaining the parameters included in Eq. (2) (Valverde-Arias et al., 2019). The robustness of this model was evaluated through the RMSE (%) and R-squared coefficient.

Specific comments on figures.

Figure 1: this figure can be merged with Figure 2 saving space (maybe Latin America map can be removed) and figure 1A is similar to figure 2 but not dividing coloured area in zones. Figure 2: see above Figure 3: the graph in figure A seems that could be fitted also linearly: how different would it be? Figure 4: images should all use the same legend, otherwise the comparison lead to wrong interpretation. Further, reducing the area to a smaller region can simplify making legend uniform. Figure 5: axis should be the same in all graphs. Figure 6: axis should be the same in all graphs.

Agree; a new combined figure has been included **(Fig. 1).**

[Figure]

**Figure 1. A) Location of Ecuador in South America, B) location of Babahoyo canton in Ecuador, and C) Agro-ecological homogeneous zones *f7* and *f15* over rice cultivation area with yield observations in Babahoyo canton**

With respect to figure 3, please see answer to reviewer's comment 5 above.

With respect to figures 4, 5 and 6, we agree with the reviewer and all figures have been changed in revised manuscript (**In revised manuscript figures 3, 4 and 5**) according to the provided comments.

Specific comments on tables.

Table 1: this table do not provide many additional information and can be removed, if you want.

Agree. The former table 1 was removed

Table 6: The use of the symbol > is a bit misleading, so I'd remove it and the unit of precipitation maybe is missing the time period.

We corrected table 6; in the new version is **table 5**. The mistaken symbol > from the altitude data was removed and we added the time period in precipitation data units.

Table 7: it seems there is some inconsistency between table and text, since Z is negative in Scenario 2 and not in Scenario 1. Further, many information are repeated in the table: would it possible to remove repetition and improve the readability?

We misplaced the scenarios in table 7, it was corrected, **table 6** in new manuscript

Table 8: the first column is a class, but it show just a single value. Maybe it would be better to show the boundary values of each class. The table could be also replaced by a graph.

We add the range and we change the table 8 by the new **figure 6**

[Figure]

**Figure 6. Frequency distribution of NDVI_ave values both observed in imagery data and estimated through GEVmim distribution in: A) *f7*, B) *f15* and C) cantonal**

Tables 9 and 10: would it be possible to merge these tables? Further why the price should be always either 371 or 371.5.

We used the official price of rice in the last three years, the price is very variable among months and years, but, if we observe it in a particular time, it is going to be the same in both zones. It means that it is not a differencing variable between these two zones.

The tables 9 and 10 were merged in the new **table 7**

**Table 7. Indemnity calculation for physiologic and economic thresholds, for each AHZ (*f7* and *f15*) and cantonal, both in scenario 1 and 2**

| | Expected Yield[*] (t/ha) | Price (USD/t) | Gross incomes (USD/ha) | Production cost scenario 1 (USD/ha)[**] | Gross margin scenario 1 (USD/ha) | Production cost scenario 2 (USD/ha)[***] | Gross margin scenario 2 (USD/ha) |
|---|---|---|---|---|---|---|---|
| *Physiologic threshold* | | | | | | | |
| *Canton al* | 2.65 | 371.50 | 984.4 | 1259 | **-274.62** | 1259 | **-274.62** |
| *f7* | 2.65 | 371.50 | 984.4 | 1022 | **-37.62** | 1259 | **-274.62** |
| *f15* | 2.65 | 371.50 | 984.4 | 1629 | **-644.62** | 1259 | **-274.62** |
| *Economic threshold* | | | | | | | |
| *Canton al* | 5.65 | 371.50 | 2099 | 1259 | **840.21** | 1259 | **840.21** |
| *f7* | 5.11 | 371.50 | 1899 | 1022 | **877.28** | 1259 | **640.28** |
| *f15* | 6.68 | 371.50 | 2482 | 1629 | **853.31** | 1259 | **1223.32** |

Table 11: it might be useful to add a column with the compensation per ha, before the total compensation for a 20 ha farm.

Agree. The suggested column was added (see the new **table 8**)

**Anonymous Referee #2**

General comments: The paper is within the scope of the journal. The paper is characterized by scientific innovation and generality, since the methodology can be applied in different regions and climatic environments. In that sense, it is extremely useful presentation of a new approach. The paper is very well and analytically presented with detailed description of the steps followed and explanation of the corresponding results.

Thank you very much for your positive comments, questions and suggestions for improvement. Below we answer to them and explain the changes made to the manuscript.

Technical-scientific comments:

1) Lines 79 and 118. In reviewing the existing remote sensing literature, it is worth to mention the system Copernicus, with Sentinel 1, or 2 or 3, which is very promising with improved resolution (20-30 m) and what is the expected improvement in the accuracy of the analysis. The authors should add comments and and/or comparisons.

We agree with the comment. We have included some examples of using Sentinel imagery in crop monitoring; and for sure, it will be an important source of data in future applications of IBI. But so far, there are not enough historical data available of Sentinel (since 2014) for being used in an IBI design. However, once we have designed the IBI application, it can be used for monitoring the crop state due to its high temporal and spatial resolution (10 days and 10-60m respectively).

The improvement of Sentinel imagery in temporal and spatial resolution with respect to MODIS imagery, could allow us to discriminate much better the extreme climatic events' impact over crops. Additionally, it could be used for IBI design for other crops that are cultivated in small and scattered parcels, as maize in Ecuador.

We explained this in **lines 129-139**:

Satellite imagery data were obtained from the MODIS MOD13Q1V6 product (NASA LP DAAC, 2015) (See Didan et al. (2015) for a description of its characteristics). MODIS imagery was selected due to its long temporal coverage (imagery data available since 2001), which is necessary for constructing a historical sequence of NDVI. MODIS's spatial resolution 250 m is moderate, but for regional applications of crop monitoring this resolution is sufficient (Jiao et al., 2019; Sánchez et al., 2018). Since 2015, Sentinel 2 imagery is available with a resolution of 10 to 60 m depending on the bands. Many studies have used Sentinel for monitoring crops state with very positive results (Inglada et al., 2015; Van Tricht et al., 2018; Veloso et al., 2017). However, current time data series availability of Sentinel does not allow its use for IBI design, as at least 10 years of historical data are needed for an insurance design (Rao, 2010). Despite this, Sentinel will become in the coming years an important alternative in the insurance field. Also, Sentinel 1, which is a radar sensor, could be an interesting option for rice monitoring in those zones where steadily presence of clouds represent a problem, as Torbick et al., (2017) mentioned in their study

2) Lines 73, 83-84 and 113. Agroecological zones. The proposed methodology uses principal component analysis (PCA) for classification. The subject of agro climatic classification and zoning is large, which considers different methodologies, including combinations of satellite indices, besides PCA. The authors should mention the main existing literature in the subject, since there might be potential improvement of the conducted agroecological zoning.

Thank you for the comment that we fully support. However, the agro-ecological zoning was the main subject in our first article (Arias et al., 2018), where our methodology was compared with other zoning methodologies. Further, this zoning was evaluated through satellite imagery for validating our resulting AHZs in a second article (Valverde-Arias et al., 2019). We did not explain this in the paper in order to focus on the IBI design. However, now we have clarified this in **lines 123-124**:

This zoning was evaluated through NDVI imagery in the study of Valverde-Arias et al., (2019), which proved that this zoning is adequate and necessary for designing an efficient IBI, reducing basis risk

3) Line 127. A clarification is required about the NDVI_ave data used: (1) these are annual values?, (2) which means that for each pixel there are 17 values? (3) How the sample points of 30% are selected? randomely?  Why only 30%?  (4) the rest are used for validation or not used at all?

NDVI_ave is the average of all NDVI measures of rice crop cycle (January to May) for each observation point, i.e. the average of the 10 NDVI measures obtained every 16 days from January to May (5 months, 2 measures per month). This means that we have one average value per year (i.e. 17 values, from 2001 to 2017) that corresponds to an average of NDVI measures along 5 months.  See **lines 140-143 and 150-151**.

With respect to the sample size, we followed the equation below (Olofsson et al., 2014), which is used for calculating the adequate sample size in a stratified random sampling of geographic products. As result of this evaluation, we obtained that the adequate stratified sample size is 12,77% of the total pixels in each AHZ. We wanted to make sure not to use a lower sample size than 12%. Thus, just for researching purposes, we decided to use 30% to ensure the representativeness of the sample. However, in our former paper in which we assessed the optimum sample size in a practical Index based insurance application, we found that even the 10% is significant representative, because the homogeneity within AHZs is very high.

The sample points were generated by an ArcGis tool that produces random sample points homogeneously distributed in the different strata (AHZs f7 and f15) in the study area. We didn't use a higher sample size or the total pixels because statistically it is not necessary, and it would increase our data processing and storage requirements. If we consider that we processed data of 17 years that increment could overcome our analysis capacity.

$$n = \frac{(\sum W_i S_i)^2}{[S(\hat{O})]^2 + (\frac{1}{N})\sum W_i S_i^2} \approx \left(\frac{\sum W_i S_i}{S(\hat{O})}\right)^2$$

Where,
$N$ is number of units in the area of interest (number of overall spatial units (pixels)
$S(O)$ is the standard error of the overall accuracy that we would like to achieve
$Wi$ is the mapped proportion of area of class $i$, and
$Si$ is the standard deviation of stratum $i$

We clarified this in **lines 152-163**

4)  Line 142. For rice estimation, what NDVI_ave values are used? annual per pixel?III.

We used the pixels' NDVI average during rice-crop cycle of rainy season (Jan-May) that corresponds temporary (same year) and spatially with the geo-referenced point of the rice-yield data. This means there is one value of NDVI_ave per year and pixel.

Technical corrections:
1.  Line 102. Never a sentence starts with number...: insteadof 84%, write:  " , where 84%...".
     It was corrected in **line 107**

2.  Line 120.  Instead "...  the precipitations", write"rainy season".
     It was corrected in **line 143**

3.  Line 277.  Instead of "Once that...", write "Since..."

4. Line 304.Never a sentence starts with number...: instead of 25%, write: " , where 25%..."

**Mentioned references**

Arias, O. V., Garrido, A., Villeta, M. and Tarquis, A. M.: Homogenisation of a soil properties map by principal component analysis to define index agricultural insurance policies, Geoderma, 311, 149–158, doi:https://doi.org/10.1016/j.geoderma.2017.01.018, 2018.

Dario, B. R.: Agricultural risk management using NDVI pasture index-based insurance for livestock producers in south west Buenos Aires province, edited by M. D. Fernando and A. P. L. P. and P. ßKen S. Tan, Agric. Financ. Rev., 75(1), 77–91, doi:10.1108/AFR-12-2014-0044, 2015.

Didan, K., Barreto, A., Solano, R. and Huete, A.: MODIS Vegetation Index User's Guide (MOD13 Series) Version 3.00, (Collection 6), , 32 [online] Available from: https://vip.arizona.edu/documents/MODIS/MODIS_VI_UsersGuide_June_2015_C6.pdf, 2015.

Inglada, J., Arias, M., Tardy, B., Hagolle, O., Valero, S., Morin, D., Dedieu, G., Sepulcre, G., Bontemps, S., Defourny, P. and Koetz, B.: Assessment of an Operational System for Crop Type Map Production Using High Temporal and Spatial Resolution Satellite Optical Imagery, Remote Sens., 7(9), 12356–12379, doi:10.3390/rs70912356, 2015.

Jiao, W., Tian, C., Chang, Q., Novick, K. A. and Wang, L.: A new multi-sensor integrated index for drought monitoring, Agric. For. Meteorol., 268, 74–85, doi:https://doi.org/10.1016/j.agrformet.2019.01.008, 2019.

Moreno, B.: Yield rice in Ecuador. First quarter 2014, Quito-Ecuador. [online] Available from: http://sinagap.agricultura.gob.ec/pdf/estudios_agroeconomicos/rendimiento_arroz_1er_cuat rimestre.pdf, 2014.

NASA LP DAAC: MOD13Q1: MODIS/Terra Vegetation Indices 16-Day L3 Global 250m Grid SIN V006, USGS Earth Resour. Obs. Sci. Center. Sioux Falls, South Dakota, doi:10.5067/MODIS/MOD13Q1.006, 2015.

Olofsson, P., Foody, G. M., Herold, M., Stehman, S. V, Woodcock, C. E. and Wulder, M. A.: Good practices for estimating area and assessing accuracy of land change, Remote Sens. Environ., 148, 42–57, doi:https://doi.org/10.1016/j.rse.2014.02.015, 2014.

Rao, K.: International Conference on Agricultural Risk and Food Security 2010, Agric. Agric. Sci. Procedia, 1, 193–203, doi:10.1016/j.aaspro.2010.09.024, 2010.

Rhee, J., Im, J. and Carbone, G. J.: Monitoring agricultural drought for arid and humid regions using multi-sensor remote sensing data, Remote Sens. Environ., 114(12), 2875–2887, doi:https://doi.org/10.1016/j.rse.2010.07.005, 2010.

Sánchez, N., González-Zamora, Á., Martínez-Fernández, J., Piles, M. and Pablos, M.: Integrated remote sensing approach to global agricultural drought monitoring, Agric. For. Meteorol., 259, 141–153, doi:https://doi.org/10.1016/j.agrformet.2018.04.022, 2018.

Torbick, N., Chowdhury, D., Salas, W. and Qi, J.: Monitoring rice agriculture across myanmar using time series Sentinel-1 assisted by Landsat-8 and PALSAR-2, Remote Sens., 9(2), doi:10.3390/rs90201019, 2017.

Van Tricht, K., Gobin, A., Gilliams, S. and Piccard, I.: Synergistic Use of Radar Sentinel-1 and Optical Sentinel-2 Imagery for Crop Mapping: A Case Study for Belgium, Remote Sens., 10(10), doi:10.3390/rs10101642, 2018.

Valverde-Arias, O., Garrido, A., Saa-Requejo, A., Carreño, F. and Tarquis, A. M.: Agro-ecological variability effects on an index-based insurance design for extreme events, Geoderma, 337, 1341–1350, doi:https://doi.org/10.1016/j.geoderma.2018.10.043, 2019.

Veloso, A., Mermoz, S., Bouvet, A., Toan, T. Le, Planells, M., Dejoux, J.-F. and Ceschia, E.: Understanding the temporal behavior of crops using Sentinel-1 and Sentinel-2-like data for agricultural applications, Remote Sens. Environ., 199, 415–426, doi:https://doi.org/10.1016/j.rse.2017.07.015, 2017.

Vrieling, A., Meroni, M., Shee, A., Mude, A. G., Woodard, J., de Bie, C. A. J. M. (Kees) and Rembold, F.: Historical extension of operational NDVI products for livestock insurance in Kenya, Int. J. Appl. Earth Obs. Geoinf., 28, 238–251, doi:https://doi.org/10.1016/j.jag.2013.12.010, 2014.

Vroege, W., Dalhaus, T. and Finger, R.: Index insurances for grasslands – A review for Europe and North-America, Agric. Syst., 168, 101–111, doi:https://doi.org/10.1016/j.agsy.2018.10.009, 2019.

Zhang, X., Chen, N., Li, J., Chen, Z. and Niyogi, D.: Multi-sensor integrated framework and index for agricultural drought monitoring, Remote Sens. Environ., 188, 141–163, doi:https://doi.org/10.1016/j.rse.2016.10.045, 2017.

**List of the relevant changes made in the revised manuscript**

It was included a new paragraph in **lines 80-84 about the reason for selecting NDVI in our study among others vegetation indexes.**

We added **lines 123-124** in the paragraph of Agro-ecological Homogeneous zones to clarify our zoning methodology and for indicating that further information can be found in (Arias et al., 2018)

It was included the new paragraph in **lines 129-139,** about the process of pixel selection taking account the pixel quality indicator of MODIS imagery. Additionally, a paragraph in which we mentioned the possible alternative to use Sentinel imagery in future Index based insurance applications.

We include the paragraph **in lines 152-163** for clarifying the sample size calculation.

It were merged the former Fig. 1 and 2 in a new Fig. 1 **(in the revised manuscript).**

In the former Figure 4, they were corrected the scale legend bar of included maps **(in revised manuscript Fig. 3).**

In Figures 5 and 6, we standardized the maximum Y axe's value, setting all of them in five (**In revised manuscript figures 4 and 5)**.

The former Table 1 was removed in the new manuscript

We corrected Table 6; in the new version is T**able 5**. The mistaken symbol > from the altitude data was removed and we added the time period in precipitation data units (mm/year).

We misplaced the scenario 1 and 2 in former Table 7. It was corrected in **Table 6** in new manuscript.

We eliminated the Table 8 and replaced it by the new F**igure 6**

The Tables 9 and 10 were merged in the new **Table 7**

We included a new column with the compensation value per hectare, in the former Table 11, now it is the **Table 8 in the new manuscript.**

.

[revised manuscript text omitted]